# Cobalt-catalyzed atroposelective C−H activation/annulation to access N−N axially chiral frameworks

Tong Li[1,2], Linlin Shi[1,2], Xinhai Wang[1], Chen Yang[1], Dandan Yang [icon][1] ✉, Mao-Ping Song[1] & Jun-Long Niu [icon][1] ✉

The N−N atropisomer, as an important and intriguing chiral system, was widely present in natural products, pharmaceutical lead compounds, and advanced material skeletons. The anisotropic structural characteristics caused by its special axial rotation have always been one of the challenges that chemists strive to overcome. Herein, we report an efficient method for the enantioselective synthesis of N−N axially chiral frameworks via a cobalt-catalyzed atroposelective C-H activation/annulation process. The reaction proceeds under mild conditions by using $Co(OAc)_2 \cdot 4H_2O$ as the catalyst with a chiral salicyloxazoline (Salox) ligand and $O_2$ as an oxidant, affording a variety of N−N axially chiral products with high yields and enantioselectivities. This protocol provides an efficient approach for the facile construction of N−N atropisomers and further expands the range of of N−N axially chiral derivatives. Additionally, under the conditions of electrocatalysis, the desired N−N axially chiral products were also successfully achieved with good to excellent efficiencies and enantioselectivities.

Atropisomerism, as one kind of intriguing axial chirality, has greatly aroused explosive attention due to their fascinating architectures and excellent multidisciplinary applications (pharmaceutical exploitation, functional materials development), arising from the restricted rotation around a single bond[1–5]. Since the pioneering work by Kenner on the expound of axial chirality from the perspective of fabrication and development significance via organic synthetic tactics in early 19th century[6], innumerable atropisomerism-based skeletons have been continually discovered due to the continuous progress in preparation, separation, purification methods and instruments. In this context, C−C and C−N atropisomers, including axially chiral biaryls, aryl amines, and aryl amides, are emerging at an amazing speed, and a variety of innovative achievements have been sparkled over the past two decades[7–19]. The atropisomerism of N−N bond, however, eluded such serious attention for a long time, which might be attributed to the incorrect notion that N−N axis is unstable due to the deplanarization effect. Indeed, the electronic barrier stemming from the repulsive interaction between the lone pairs on the two nitrogen atoms can strongly favor the formation of such atropisomers. These intriguing atropisomerism configurations have moved onto the research stage again due to the new structural understanding, making them a superior platform for serving N−N axial natural products, bioactive molecules, functional materials, and ligands production (Fig. 1a)[20–25]. By contrast, the atropisomerism bearing a N−N bond remains largely underdeveloped, possibly due to the shorter length and weaker nature of the N−N bond[26]. Besides, the low rotational barrier caused by the two N-containing planes deplanarization, renders the N−N axis construction more challenging. Typically, the N−N axially chiral skeletons could be built up through four types of transformations (Fig. 1b). The atroposelective N−H functionalization, such as allylation, alkylation, and acylation, represents an effective mean by modifying the central N−N axis[27–30]. Alternatively, catalytic asymmetric desymmetrization of pro-chiral substrates[31,32], and asymmetric assembly via de novo construction of pyrroles and indoles[33–37] also provide satisfactory choices

[1]College of Chemistry, Zhengzhou University, Zhengzhou 450001, P. R. China. [2]These authors contributed equally: Tong Li, Linlin Shi. ✉e-mail: yangdandan@zzu.edu.cn; niujunlong@zzu.edu.cn

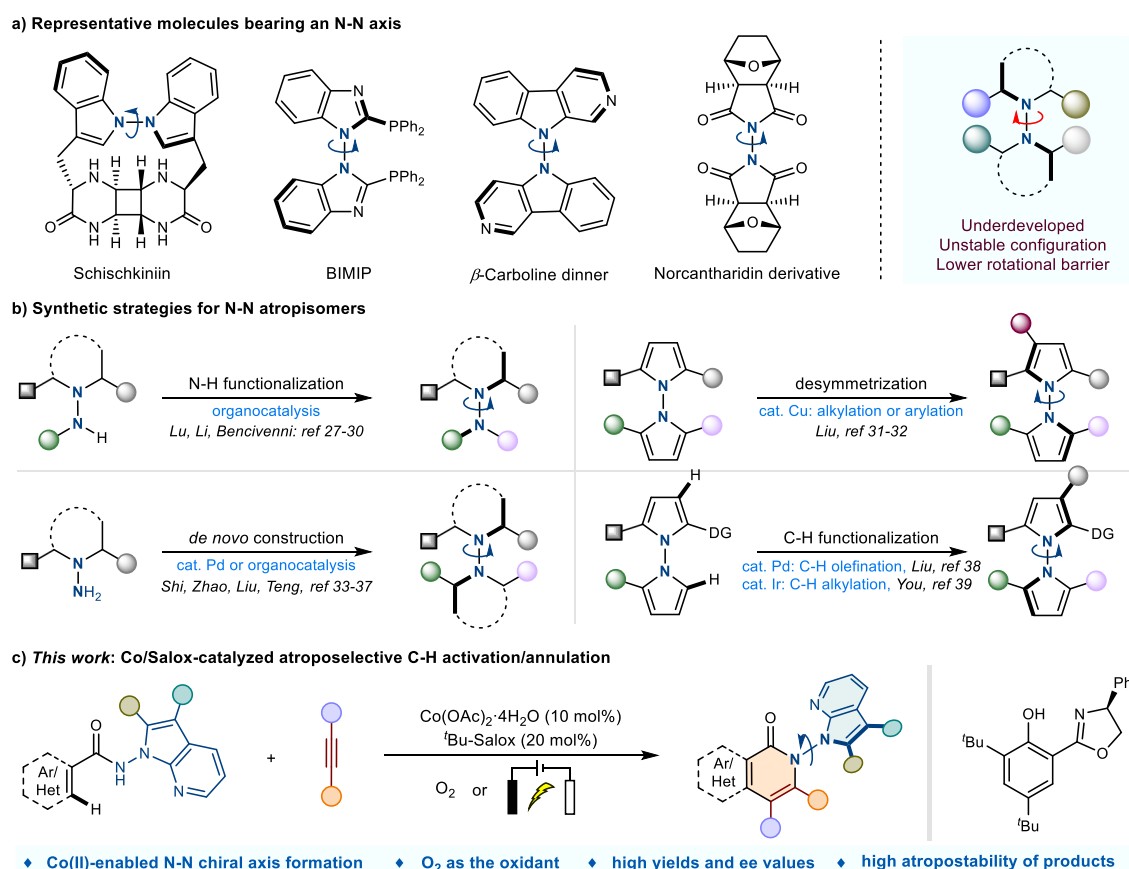

**Fig. 1 | Background and project synopsis. a** Representative molecules bearing an N−N axis. **b** Synthetic strategies for N−N atropisomers. **c** This work: Co/Salox-catalyzed atroposelective C–H activation/annulation.

for creating N−N atropisomers. Typically, Shi group have recently achieved organocatalytic enantioselective construction of N−N axially chiral indoles and pyrroles[36,37]. More recently, Liu and You groups successfully reported the construction of N−N atropisomers via the Pd or Ir-catalyzed C−H functionalization of indole/pyrrole ring through dynamic kinetic resolution of racemic N−N biaryls[38,39], further advancing the development of this field. Despite the elegant efforts made, research in this area is still in its infancy and remains underdeveloped. As a result, the development of more efficient and facile approaches for building novel and diverse N−N atroposelective frameworks is highly appealing yet challenging.

For this purpose, asymmetric C−H functionalization, as a timely emerged and well-established tactic, was built up to form diverse charming structures containing highly functionalized skeletons, such as natural products, pharmaceutical lead compounds, important architectures which typically takes advantage of batch production with satisfactory chemical selectivity (stereoelectronic selectivity and regioselectivity) via transition metal catalyzed process[40–45]. In this content, the first-row 3d-metal cobalt-catalyzed C−H functionalization has gained increasing popularity owing to its earth-abundance, low toxicity, and distinctive reactivities[46–53], which has been largely developed for the synthesis of various skeletons by Ackermann[54–56], Yoshino and Matsunaga[57–60], Cramer[61–63], Shi[64–69], and our groups[70–73]. Very recently, our group[70], and Shi's group[65] independently developed an efficient chiral Co/Salox (salicyl-oxazoline) system, and demonstrated its excellent reactivity for assembling C−N and C−C axially chiral compounds. This strategy features significant advantages, making easily available cobalt salt catalyzed C−H functionalization for constructing axial chiral compounds being a promising research area. On the other hand, the nitrogen atoms in the N−N atropisomers may be derived from different substructures, giving rise to the possibility of

forming diverse N−N axially chiral compounds. However, to the best of our knowledge, the construction of N−N axially chiral atropisomers via earth abundant transition-metal catalyzed C−H activation remains unprecedented.

Inspired by the elegant and well-established cobalt-catalyzed C−H activation/functionalization developments, and with our continuing interest in cobalt catalysis, we hope to achieve further development in this field through the strategy of direct asymmetric C−H activation. Delightedly, we here reported a highly efficient example of cobalt-catalyzed atroposelective construction of N−N axially chiral frameworks, combining the advantages of excellent efficiency and enatioselectivities (up to 98% yield and 99% e.e.) based on cobalt/Salox catalysis, which undoubtfully broaden the synthetic strategies for the direct and effective construction of N−N atropisomers and further enrich the types/scopes of N−N axially chiral derivatives. This approach has proved to be a highly suitable way for efficient preparation of N−N atropisomers according to their readily available raw materials, and accessible operation characteristics (Fig. 1c). Notable features of this protocol include: (1) the cheap cobalt(II) salt enabled N−N chiral axis construction, (2) the use of environmentally friendly $O_2$ as the oxidant, (3) high yields and excellent enantioselectivities with good functional group tolerance, (4) successful application under electrochemical conditions, (5) the unique atropostability for the N−N axially chiral products.

## Results
### Optimizing reaction conditions
To validate the feasibility of the hypothesis, our investigation was initiated by using the *N*-(7-azaindole)benzamide **1a** and phenylacetylene **2a** as the model substrates, with 10 mol% of Co(OAc)$_2$·4H$_2$O as the catalyst and $O_2$ as the oxidant (Table 1). Diverse chiral Salox ligands

**Table 1 | Optimization studies[a]**

Co(OAc)$_2$·4H$_2$O (10 mol%), L (20 mol%), O$_2$, additive, solvent, T, t

**L1** 15%, 99% ee

**L2**, R = Bn, N.R — **L3**, R = $^t$Bu, N.R

**L4**, R = OMe, 15%, 99% ee — **L5**, R = F, 5%, 99% ee

**L6** 48%, 99% ee

**L7** 42%, 98% ee

| Entry | Ligand | Co catalyst | Additive | Solvent | T (°C) | Time | Yield (%) | ee (%) |
|---|---|---|---|---|---|---|---|---|
| 1 | L6 | Co(OAc)$_2$·4H$_2$O | – | Toluene | 100 | 6 | 48 | 99 |
| 2 | L6 | Co(OAc)$_2$ | – | Toluene | 100 | 6 | 48 | 98 |
| 3 | L6 | Co(OBz)$_2$ | – | Toluene | 100 | 6 | 45 | 98 |
| 4 | L6 | CoCl$_2$·6H$_2$O | – | Toluene | 100 | 6 | 0 | - |
| 5 | L6 | Co(OAc)$_2$·4H$_2$O | – | 1,4-Dioxane | 100 | 6 | 68 | 98 |
| 6 | L6 | Co(OAc)$_2$·4H$_2$O | – | THF | 100 | 6 | 42 | 98 |
| 7 | L6 | Co(OAc)$_2$·4H$_2$O | – | DME | 100 | 6 | 43 | 98 |
| 8 | L6 | Co(OAc)$_2$·4H$_2$O | – | CPME | 100 | 6 | 57 | 98 |
| 9 | L6 | Co(OAc)$_2$·4H$_2$O | NaOPiv·H$_2$O | 1,4-Dioxane | 100 | 6 | 72 | 98 |
| 10 | L6 | Co(OAc)$_2$·4H$_2$O | AcOH | 1,4-Dioxane | 100 | 6 | 69 | 98 |
| 11 | L6 | Co(OAc)$_2$·4H$_2$O | AdCO$_2$H | 1,4-Dioxane | 100 | 6 | 94 | 98 |
| 12 | L6 | Co(OAc)$_2$·4H$_2$O | AdCO$_2$H | 1,4-Dioxane | 80 | 6 | 94 | 98[b] |
| 13 | L6 | Co(OAc)$_2$·4H$_2$O | AdCO$_2$H | 1,4-Dioxane | 80 | 4 | 89 | 98 |

ee enantiomeric excess, N.R. no reaction, THF tetrahydrofuran, DME 1,2-dimethoxyethane, CPME cyclopentyl methyl ether.
[a]Unless otherwise mentioned, all reactions were carried out using **1a** (0.1 mmol), **2a** (0.12 mmol), L6 (0.02 mmol), Co(OAc)$_2$·4H$_2$O (0.01 mmol) in toluene (1 mL) at 100 °C under O$_2$ for 6 h, isolated yields.
[b]90% yield and 98% ee for 0.2 mmol scale.

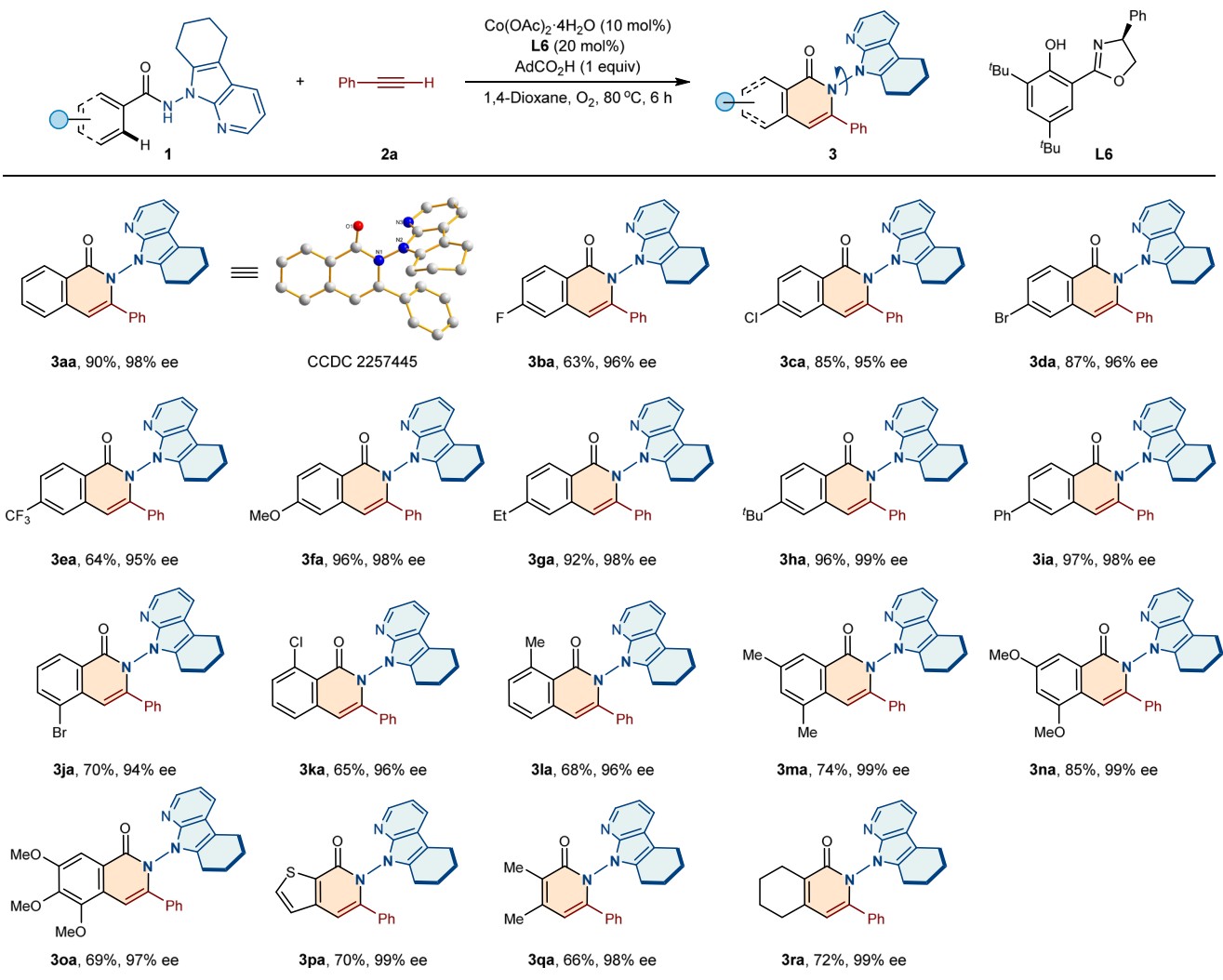

**Fig. 2 | Scope of benzamides.** Reaction conditions: **1** (0.2 mmol), **2a** (0.24 mmol), Co(OAc)$_2$·4H$_2$O (0.02 mmol), **L6** (0.04 mmol), AdCO$_2$H (0.2 mmol) in 1,4-dioxane (2 mL) at 80 °C under O$_2$ for 6 h, isolated yields. ee, enantiomeric excess.

**L1**–**L7** were evaluated. It was pleased to find that the desired N–N axially chiral isoquinolinone product **3aa** was obtained with 48% yield and excellent 99% ee, with **L6** bearing bulky *tert*-butyl groups at the *ortho*- and *para*-position of phenol as the chiral ligand. And this promoting effect of the bulky *t*Bu-Salox ligand has also been reported in the previous Co/Salox catalysis developed by Shi[69], Ackermann[56] and our group[73]. Various cobalt catalysts, such as Co(OAc)$_2$, and Co(OBz)$_2$, CoCl$_2$·6H$_2$O, were fully investigated, in which no better results have got (entries 1–4, and Supplementary Table 2). The effects of the solvent were then screened, and 1,4-dioxane displayed the best activity (entries 5–8, and Supplementary Table 3). Further optimization was conducted by the use of additives, such as NaOPiv·H$_2$O, AcOH, and AdCO$_2$H (entries 9–11, and Supplementary Table 4), and AdCO$_2$H proved to be the best choice. Further optimization of the reactive temperature, time, and catalyst/ligand loadings, revealed that the reaction was able to produce the atroposelective isoquinolinone **3aa** with 94% yield and 98% ee at 80 °C for 6 h (entries 12–13, and Supplementary Tables 5–8). The absolute configuration of N–N axially chiral **3aa** was determined to be *R*$_a$ by X-ray diffraction analysis (CCDC 2257445). In addition, a competitive experiment by using **1a** and **1a'** bearing a 7-methyl-8-aminoquinoline directing group with a 1:1 ratio as substrate was conducted under optimized conditions, and the product **3aa** and **3aa'** were obtained with a total yield of 99% with a 1:1 ratio (See Supplementary Table 10).

## Substrate scope

With the optimal conditions in hand, we then explored the substrate generality of this transformation (Fig. 2). A variety of benzamides with diverse functionalities at *para*-position were systematically investigated. To our delight, both electron-withdrawing (−F, −Cl, −Br and −CF$_3$) and electron-donating (−OMe, −Et, −*t*Bu, and −Ph) groups were well tolerated to afford the N–N axially chiral isoquinolinones **3ba–3ia** in good yields (63–97%) and excellent enantioselectivities (95–99% ee). For the *meta*-substituted substrate **1j**, the annulation took place at the sterically more hindered position to provide **3ja** in 70% yield and 94% ee. *Ortho*-substituted benzamides **1k** and **1l** also reacted smoothly, delivering **3ka** and **3la** with good yields and high enantiopurities of 96% ee. When multi-functionalized substrates with methyl or methoxy substituents located at the aromatic ring, the desired N–N axially chiral isoquinolinones **3ma–3oa** could also be obtained with good to excellent efficiencies and enantioselectivities. Notably, the heterocyclic substrate **1p** containing a thienyl moiety also reacted smoothly to give the product **3pa** with 70% yield and 99% ee. In addition, the challenging vinylamides **1q** and **1r** were well compatible with this atroposelective transformation, delivering products **3qa** and **3ra** with high level of enantiocontrol (98–99% ee).

Alkyne substrates were further expanded to verify the generality of the protocol. As shown in Fig. 3, both electron-donating (−OMe) and electron-withdrawing (−CF$_3$) groups at the *para*-position of

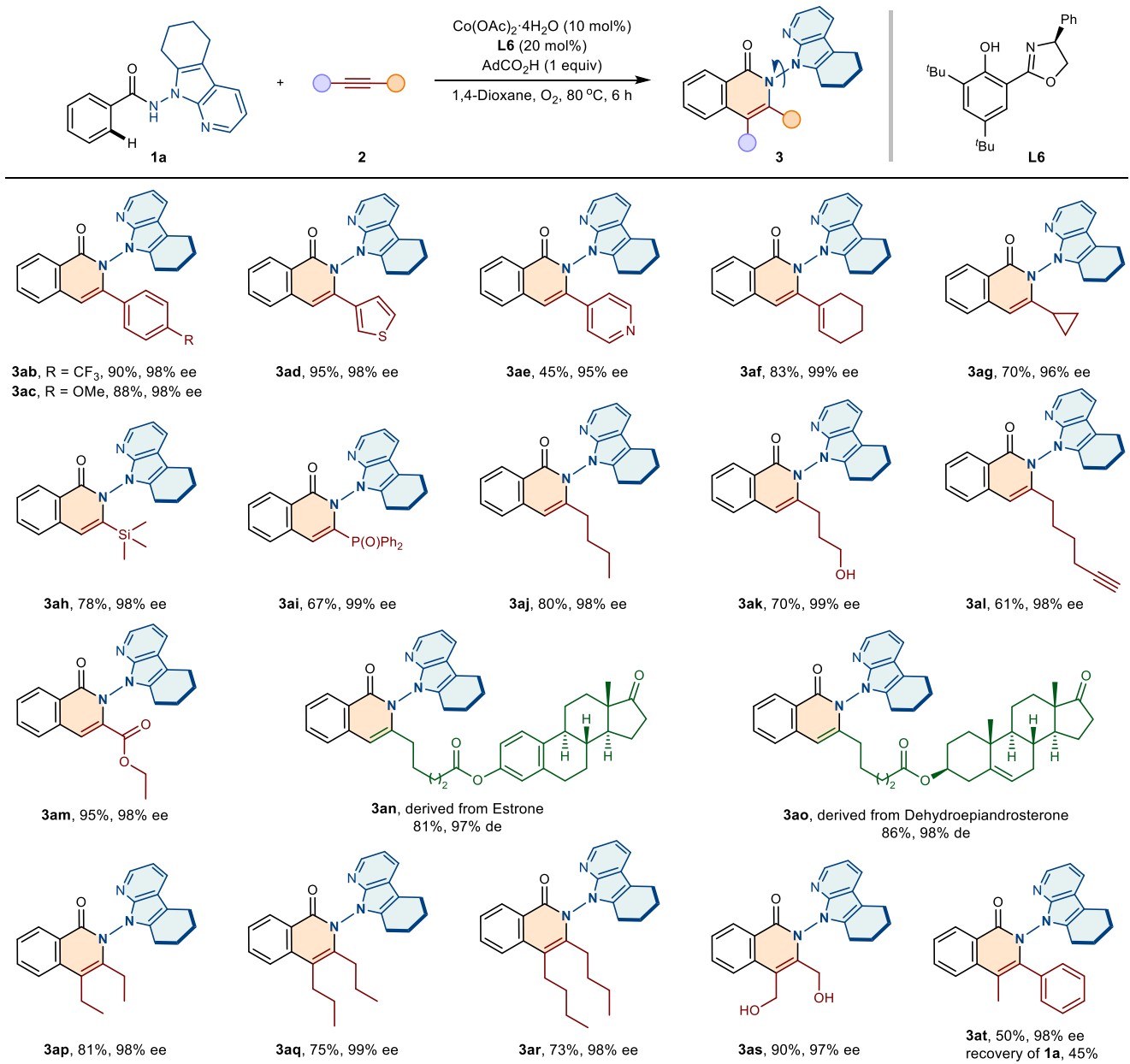

**Fig. 3 | Scope of alkynes.** Reaction conditions: **1a** (0.2 mmol), **2** (0.24 mmol), Co(OAc)$_2$·4H$_2$O (0.02 mmol), **L6** (0.04 mmol), AdCO$_2$H (0.2 mmol) in 1,4-dioxane (2 mL) at 80 °C under O$_2$ for 6 h, isolated yields. de, diastereomeric excess.

phenylacetylene were well tolerated, and the corresponding products **3ab** and **3ac** were obtained in 90 yield, 98% ee and 88% yield, 98% ee, respectively. The heterocycle-substituted alkynes (including 3-ethynylthiophene and 4-ethynylpyridine), conjugated enyne, and cyclopropyl acetylene were also found to be compatible with this reaction, furnishing the desired products **3ad**–**3ag** with excellent enetioselectivities (95–99% ee). Notably, the presences of Si and P substituents in the resulting N–N axially chiral products (**3ah**–**3ai**) were particularly important for further derivation owing to their inherent reactivity. In addition, the alkyl or ester-substituted terminal alkynes, even with functional groups, proved to be suitable coupling partners to give products **3aj**–**3am** with good yields (61–95%) and high enantiopurities (98–99%). To further demonstrate the practicality of the protocol, the natural product-linked alkynes **2n** and **2o** were synthesized and utilized for the related atroposelective annulation, and the target products were successfully obtained in 81–86% yields and high diastereomeric excess (97–98% de). Moreover, the aliphatic

internal alkynes **2p**–**2s** also reacted smoothly to deliver N–N axially chiral isoquinolinones **3ap**–**3as** with 97–99% ee values. For the unsymmetrical internal alkyne **2t**, the reaction proceeded with exclusive regioselectivity to give **3at** in moderate yield and high enantioselectivity, and the starting material of **1a** was recovered with a 45% yield. But the diphenylacetylene was incompatible in this system, possibly due to the steric hindrance.

Subsequently, the scope of the benzamides bearing different substituents on the 7-azaindole ring were investigated. As shown in Fig. 4, for azaindole backbone, which was functionalized by cyclohexane moiety, was well compatible, generating **3sa** in good yield (85%) and high enantiopurity (97% ee). The protocol also tolerated 2-ethyl and 2-methyl substituted azaindoles, furnishing the desired products **3ta** and **3ua** with high enantiocontrol (98% ee and 97% ee, respectively). In addition, benzamides bearing functionalities located on the C2 and C3 position of the azaindole ring, reacted smoothly to deliver N–N axially chiral products **3va**-**3xa**, with good

reactivities (70–81% yields) and excellent enantioselectivities (97–98 % ee).

To gain more efficiencies into the reactive practicality, related electrochemical experiments were conducted (Fig. 5). The general substrates were also compatible to afford the corresponding N–N axially chiral isoquinolinones **3aa** and **3ia** without any reduction of enantioselectivities. For silicon and dialkyl substituents based on alkynes, the current protocol also proved to be a powerful tactic for diverse N–N axially chiral frameworks (**3ah** and **3aq**).

### Study on product stability and synthetic applications

To further explore the conformational stability of N–N axially chiral isoquinolinones, the racemization experiments were investigated (Fig. 6). Initially, the newly formed **3aa** was heated up to 160 °C in dodecane, and the ee value can be maintained without decreasing. The ee value of **3aa** decreased by 0.53% after 10 h at 180 °C. And the rotational energy barrier of **3aa** was calculated to be 41.6 kcal/mol, and its half-life ($t_{1/2}$) was up to $5.5 \times 10^9$ years at 25 °C. As the experimental results shown, energy barrier and the $t_{1/2}$ of **3aq** and **3ua** were also

exhibited, indicating that the N–N axially chiral isoquinolinones endowed a high degree of atropostabilities. It is worth noting that the compounds **3aa**, **3aq**, and **3ua** underwent slight decomposition after being heated for 6 or 10 h. And the recovery rates for **3aa**, **3aq**, and **3ua** were 70%, 60%, and 90%, respectively. To verify the accuracy of the testing rotational energy barriers, we further studied the enantiomerization processes of **3aa** and **3ua** by DFT calculations. As shown in Fig. 6 and Supplementary Fig. 6, the energy barriers of the dihedral rotation via transition states **TS**[3aa] and **TS**[3ua] are 40.3 kcal/mol and 41.2 kcal/mol respectively, which showed good agreement with the experimental results.

The gram-scale experiment using **1a** (3.5 mmol, 1.02 g) with **2a** under standard conditions delivered **3aa** without obvious decrease in yield and enantioselectivity (88% yield, 96% ee). Furthermore, the annulation of **1a** with diphenylphosphinoacetylene (**2i**) on a gram-scale was caried out, leading to the formation of phosphoryl product **3ai** with 60% yield and 99% ee (Fig. 7a). The synthetic utility of this developed protocol to access N–N atropisomers was also further highlighted by post-functionalization. The compound **3ai** was reduced to produce monophosphine product **4** with a remarkable 99% ee. In

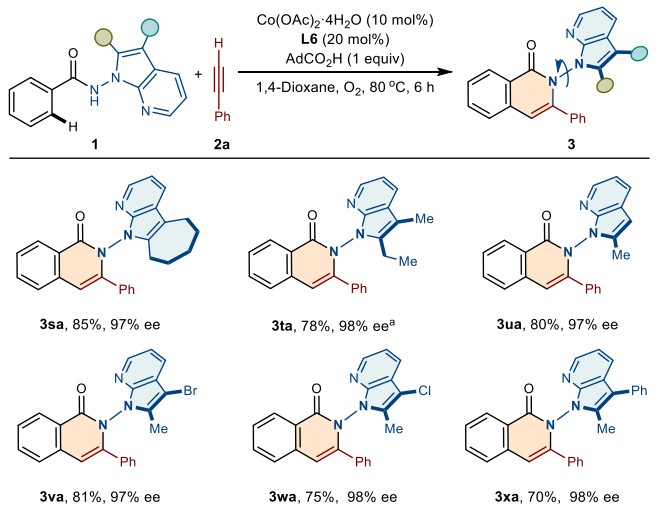

**Fig. 4 | Scope of Benzamides with Substituents on the 7-Azaindole Ring.** Reaction conditions: **1** (0.2 mmol), **2a** (0.24 mmol), Co(OAc)₂·4H₂O (0.02 mmol), **L6** (0.04 mmol), AdCO₂H (0.2 mmol) in 1,4-dioxane (2 mL) at 80 °C under O₂ for 6 h, isolated yields. ᵃ20 mol% Co(OAc)₂·4H₂O, 10 h. ee enantiomeric excess.

**Fig. 6 | Study on product stability.** The rotation barrier and the $t_{1/2}$ of **3aa**, **3aq**, and **3ua**.

**Fig. 5 | Reaction scope for the electro-oxidative annulation.** Reaction conditions: graphite felt anode, Pt-plate cathode, constant current = 2 mA, **1** (0.2 mmol), **2** (0.24 mmol), Co(OAc)₂·4H₂O (0.02 mmol), **L7** (0.04 mmol), TFE (5 mL), NaOPiv·H₂O (0.4 mmol), 60 °C, 6 h, air, isolated yields. ee, enantiomeric excess; TFE, 2,2,2-trifluoroethanol.

**Fig. 7 | Gram-scale experiments and synthetic applications. a** Gram-scale experiments using **1a** as substrates and post-functionalization of N−N axially chiral product **3aa** and **3ai**. ee, enantiomeric excess; DDQ, 2,3-dichloro-5,6-dicyano-1,4-benzoquinone; *m*-CPBA, 3-chloroperoxybenzoic acid; BSA, *N,O*-bis(trimethylsilyl)-acetamide; DCM, dichloromethane; THF, tetrahydrofuran; rt, roomtemperature. **b** Applications of N−N axially chiral compound **4** and **6** as chiral ligands for further applications.

addition, the oxidative product **5** could be obtained by treatment of N-N axially chiral isoquinolinone **3aa** with DDQ in 88% yield at 120 °C for 2 h. Further treatment of **5** with *m*-CPBA afforded *N*-oxide **6**. As the experimental result shown, the carbonyl group of N−N axially chiral isoquinolinone could also be easily transformed into a thiocarbonyl group in 75% yield without compromising enantioselectivity at the presence of Lawesson's reagent (Fig. 7a). Monophosphine product **4** could serve as a suitable chiral ligand for the asymmetric Pd-catalyzed allylic substitution reaction of indole and the Tsuji-Trost reaction, and the desired products were obtained with excellent yields and good to high enantioselectivities (73% ee for **8**, 93% ee for **9**). In addition, *N*-

oxide **6** was regarded as the chiral ligand to react with benzyl 2-diazo-2-phenylacetate and 1,2-dimethyl-1*H*-indole in the presence of palladium catalyst to give desired **10** in 90% yield and 58% ee value (Fig. 7b)[70].

To investigate the reaction mechanism, deuterium labeling experiments were conducted. H/D exchange experiments illustrated that the C-H cleavage was irreversible (Fig. 8a). Then, the parallel and competitive kinetic isotope effect (KIE) values of 1.7 and 1.9 were calculated, indicating that C-H cleavage may be involved in the rate-determining step (Fig. 8b). Furthermore, the study on the nonlinear effect between the enantiomeric excess (ee) of product **3aa** and the ee of **L6** revealed that a single chiral ligand coordinated with a cobalt

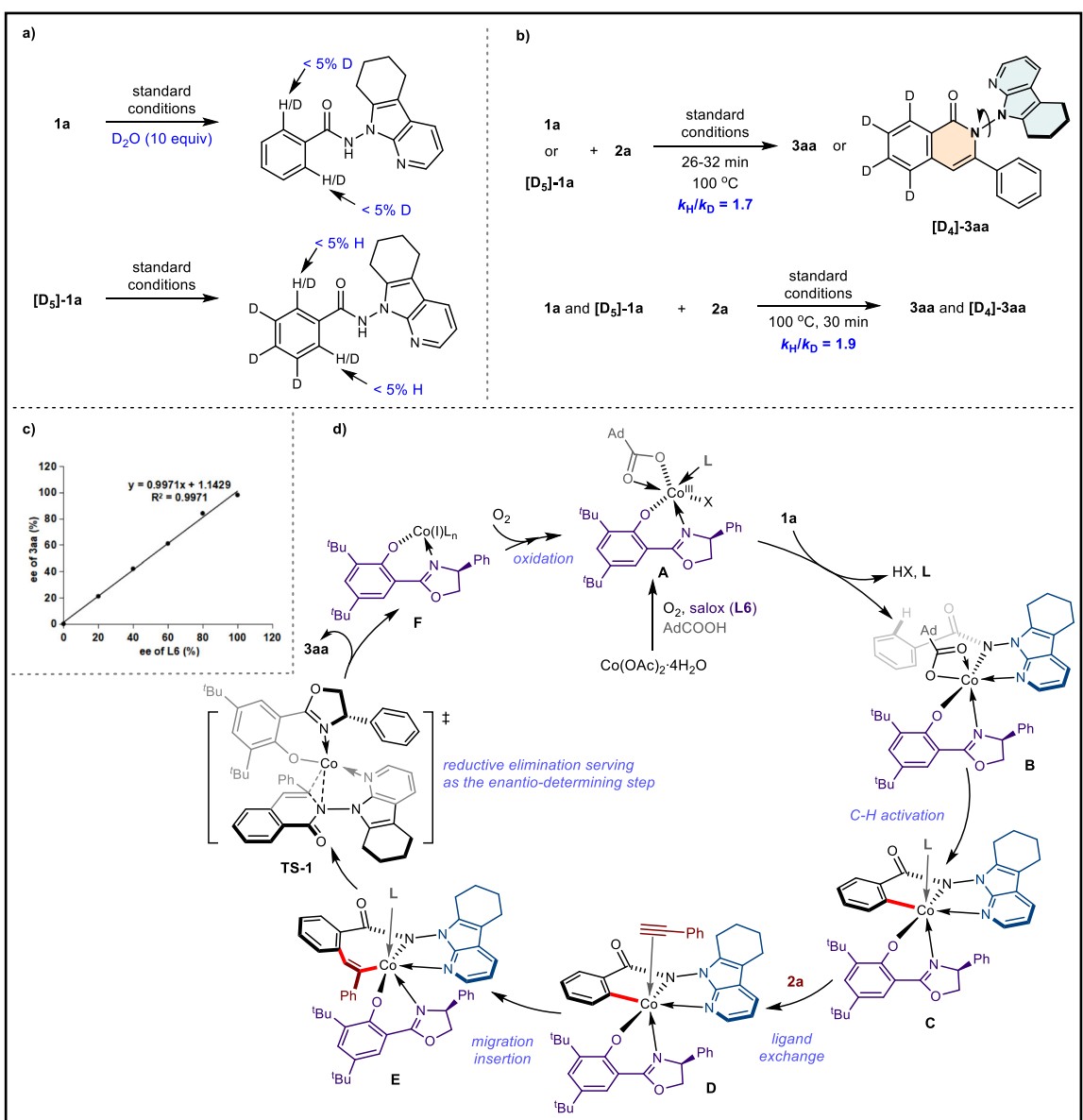

**Fig. 8 | Mechanistic studies. a** H/D Exchange experiments using **1a** or **[D₅]−1a** as substrates. **b** Parallel and competitive kinetic isotope experiments to calculate KIE values. **c** Nolinear effect study between the enantiomeric excess (ee) of product **3aa** and the ee of **L6**. **d** Plausible mechanism for the Co/Salox catalyzed annulation.

atom to create an effective catalyst (Fig. 8c). Based on the previous reports[65–73] and the above results, a plausible mechanism was proposed (Fig. 8d). Cobalt salt (II) coordinates with **L6** and AdCO₂H, and undergoes oxidation with O₂ to produce active Co(III)-species **A**, which then undergoes ligand exchange with substrate **1a** to form the octahedral Co(III)-intermediate **B**. The C–H activation of **B** affords the key intermediate **C**. It is worth mentioning that another pathway involving the first C-H activation step and the following oxidation step to form the intermediate **C** could not be completely ruled out. Next, alkyne **2a** coordinates with **C** and the ligand exchange occurred to form **D**. The migration insertion of alkyne into the C-Co bond affords the seven-membered alkenyl intermediate **E**. Finally, the reductive elimination of **E** leads to the formation of the N−N axially chiral product **3aa** and meanwhile releases the Co(I) intermediate **F**. Based on our previous report for the synthesis of C−N axially chiral compounds[70], the reductive elimination through transition states **TS-1**, which features the lowest-energy owing to the π-π stacking interactions between phenyl group of **L6** with the 7-azaindole directing group of **1a**, might serve as the enantio-determining step.

In conclusion, we have developed an efficient and facile synthetic method for accessing N−N axially chiral isoquinolinones through cobalt-catalyzed atroposelective C−H activation/annulation. This protocol exhibits several unique characteristics, including a broad substrate scope, environmentally friendly O₂ as the oxidant, and excellent efficiencies and enantioselectivities. The obtained N−N axially chiral isoquinolinones features unique atropostability, which further expands the repertoire of N−N axially chiral derivatives. In addition, this method has proven to be a powerful strategy for constructing the N−N axial architectures under electrochemical conditions for asymmetric cobalt/Salox catalysis. Furthermore, other types of N−N axially chiral compounds, and related applications to asymmetric catalysis, functional materials are being evaluated in our laboratory and will be reported in due course.

## Methods
### General procedure for the synthesis of compounds 3
An oven dried schleck tube charged with magnetic stirrer added benzamide/vinylamide **1** (0.2 mmol), Co(OAc)₂·4H₂O (0.02 mmol,

10 mol%), **L6** (0.04 mmol, 20 mol%), 1-adamantanecarboxylic acid (0.2 mmol, 1.0 equiv) with subsequent addition of 1,4-dioxane (2 mL) as solvent. To this reaction mixture, **2** (0.24 mmol, 1.2 eq) was added under $O_2$. Then, the reaction system was stirred at 80 °C for 6 h. After the reaction was completed, the reaction mixture was quenched with saturated $NaHCO_3$ solution and extracted with $CH_2Cl_2$. The combined organic layer extracts were washed with brine, dried over $Na_2SO_4$, and concentrated under reduced pressure, and purified on silica gel chromatography (petroleum ether/ethyl acetate = 5:1) to afford the corresponding products. Full experimental details and characterization of the compounds are given in the Supplementary Information.

## Data availability
The X-ray crystallographic coordinates for structures reported in this study have been deposited at the Cambridge Crystallographic Data Centre (CCDC), under deposition number 2257445 (for **3aa**). These data can be obtained free of charge from The Cambridge Crystallographic Data Centre via www.ccdc.cam.ac.uk/data_request/cif. The authors declare that the data supporting the findings of this study are available within the article and its Supplementary Information Files. Cartesian coordinates of the calculated structures are available in Supplementary Data 1. All other data are available from the corresponding authors upon request.

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

## Acknowledgements

We thank for the support of the National Natural Science Foundation of China (22271260 to J.-L.N.; 22101267 to L.S.), and Natural Science Foundation of Henan Province (222300420291 to D.Y.).

## Author contributions

J.-L.N., D.Y., and L.S. conceived the concept and prepared the manuscript. T.L., X.W., and C.Y. conducted the experiments and analyzed the data. M.-P.S. provided revisions. All the authors participated in the

discussion and preparation of the manuscript. J.-L.N., D.Y., and L.S. directed the project.

## Competing interests
The authors declare no competing interests.
