## [Peer Review File · Nature Communications]

Cobalt-catalyzed atroposelective C–H activation/annulation to access N–N axially chiral frameworksReviewers' Comments:

Reviewer #1:

Remarks to the Author:

In this manuscript, Niu and coworkers reported the catalytic synthesis of N–N axially chiral frameworks via a cobalt-catalyzed atroposelective C-H activation/annulation process. The reaction proceeds under mild conditions by using Co(OAc)₂·4H₂O catalyst with a chiral Salox ligand and O₂ as an oxidant, affording a variety of N–N axially chiral products in good yields with excellent enantioselectivities. Additionally, this method has also proven to be a powerful strategy for constructing the N–N axial architectures under electrochemical conditions for asymmetric cobalt/Salox catalysis. It is a topic of interest to the researchers in the atropisomeric areas and I would like to recommend it for publication in Nature Communications after minor revisions :

1. In line 13, "the first catalytic synthesis of N–N axially chiral frameworks has been accomplished" is not accurate, the position of attributive here is wrong.
2. It is weird to say "The Lu, and Li groups have made significant contributions to this field." Is the work done by others not significant?
3. To explore the conformational stability of N–N axially chiral isoquinolinones, the racemization experiments were investigated by heating the substrate at 180 °C. Is the product not decomposed here? The author needs to indicate the recovery rate of the product. Here it is best to calculate the rotational energy barrier of this class of compounds by DFT calculation.
4. In line 153, the production rate of 3as is surprisingly high. Is there any effect of hydroxyl group here? Can the author give a reasonable explanation.
5. A related publication Angew. Chem. Int. Ed. 2023, 62, e202216863 should be added to the references.
6. Are the unsymmetric non-terminal alkynes involved in the reaction regioselective? For example, how about the regioselectivity in 3at?

Reviewer #2:

Author reported a cobalt-catalyzed atroposelective C-H activation/annulation to synthesize N–N axially chiral isoquinolinones under electrochemical conditions. This method exhibits wide substrate scope and excellent enantioselectivities. The novelty of protocol meets the publication request for the nature communication. However, some further revisions are still needed.

- 1, Author should provide more applications to display the usefulness of the products instead of just providing a special example.
2. Although author made some experiments to testify the C-H activation may be involved in the rate-determining step, how the chirality is controlled? what is the favorable model?
3. Reaction condition screening showed that AdCO₂H is very important for the yield improvement, author should provide the complete figure for the mechanism to exhibit the usefulness of AdCO₂H and the other key intermediates.

Reviewer #3:

Remarks to the Author:

The manuscript by Niu and coworkers reports on an enantioselective annulation reaction between benzamide derivatives with azaindole-based N,N-bidentate directing group and alkynes for the synthesis of N-N atropisomeric compounds. The reaction is promoted by a cobalt-chiral salicyloxazoline (Salox) catalyst to give biheteroaryls featuring N-N bond between isoquinolinone and azaindole in good yields with excellent enantioselectivities. The reaction tolerates a broad range of terminal and internal alkynes and substituted benzamide/acrylamide derivatives. A variation in the azaindole moiety is also demonstrated. The reaction is performed under attractive aerobic conditions using O₂ as the sole oxidant, while the feasibility of electrochemical reaction setup is also shown. High atropostability of the N-N products is demonstrated. The tolerance to phosphorus-substituted alkynes allowed the authors to use the product to prepare a chiral phosphine ligand for asymmetric catalysis. Overall, this work is another remarkable example of the cobalt/Salox catalytic systems for highly enantioselective C-H transformations, which has been pioneered by the authors and the Shi group. On one hand, this may be regarded as a logical extension of the earlier study by the authors on Co/Salox-catalyzed annulation to form C-N atropisomeric products. On the other hand, the transformation and the products can be considered novel in the context of the rapidly emerging chemistry of N-N atropisomerism. Taking these into consideration, this reviewer would decide that the present work deserves publication in Nature Communications with some revision.

1. In the abstract, the authors wrote "Herein, the first catalytic synthesis of N-N axially chiral frameworks has been accomplished via...". This sentence is ambiguous and misleading. The authors should more specifically elaborate what aspects of this work are different from the previous catalytic asymmetric synthesis of N-N atropisomers.

2. The present reaction is different from the previous annulation using 8-aminoquinoline-based directing group with respect to the skeleton of key cyclometalated intermediate (5,5-fused ring for this work and 5,6-fused ring for the previous work). Concerning this difference, the authors are suggested to address the following points: 1) Which is the better directing group for this annulation? A competition reaction under both the previous and present conditions would be informative. 2) The salox ligands with the phenyloxazoline moiety appear to show uniformly good enantioselectivity for this and previous reactions regardless of the presence of the bulky tert-butyl group (L1 vs L6, L7). Thus, the tert-Bu group seems to be increasing the catalytic activity while not particularly influence the enantioselectivity. What would be behind this effect? A discussion, or even insights from computational study would be worthwhile.

3. The way of writing in this manuscript is overly superfluous. For example, the first sentence of the main text, "The natural evolution of ecosystems for centuries..." has too little connection to what is actually achieved here. Also, the use of words like "bright future" in page 2 is too fluffy and objective and thus should be avoided. The authors should be aware of the scope/limitation of implications and impacts of their actual results and adopt a solid and subjective writing style. The present results are interesting enough without superfluous writing.

Re: Reviewer #1

We appreciate a lot for the generous support from reviewer #1 on our work: “It is a topic of interest to the researchers in the atropisomeric areas and I would like to recommend it for publication in Nature Communications after minor revisions”.

Original comments: (1) In line 13, “the first catalytic synthesis of N–N axially chiral frameworks has been accomplished” is not accurate, the position of attributive here is wrong.

Response: We thank reviewer 1 for this professional suggestion. According to the suggestion, we have changed the description as “*Herein, we report an efficient method for the enantioselective synthesis of N–N axially chiral frameworks via a cobalt-catalyzed atroposelective C-H activation/annulation process.*” in the revised manuscript. And the revisions are highlighted in green background.

(2) It is weird to say “The Lu, and Li groups have made significant contributions to this field.” Is the work done by others not significant?

Response: Thanks for the considerate comments. We have deleted the inappropriate description in the instruction part.

(3) To explore the conformational stability of N–N axially chiral isoquinolinones, the racemization experiments were investigated by heating the substrate at 180 °C. Is the product not decomposed here? The author needs to indicate the recovery rate of the product. Here it is best to calculate the rotational energy barrier of this class of compounds by DFT calculation.

Response: Thanks for this professional comment. We fully understand this concern and appreciate the reviewer’s comments. First, we isolated the recovery compounds 3aa, 3aq, and 3ua after conducting the racemization experiments at 180 °C and recorded the recovery rates of these

compounds. Additionally, for the racemization processes of 3aa and 3ua, the energy barriers of the dihedral rotation have been studied by DFT calculation. All the results were added in the revised manuscript and SI, as shown below: “It is worth noting that the compounds 3aa, 3aq, and 3ua underwent slight decomposition after being heated for 6 or 10 hours. And the recovery rates for 3aa, 3aq, and 3ua were 70%, 60%, and 90%, respectively. To verify the accuracy of the testing rotational energy barriers, we further studied the enantiomerization processes of 3aa and 3ua by DFT calculations. As shown in Fig. 2 and Figure S6, the energy barriers of the dihedral rotation via transition states TS^{3aa} and TS^{3ua} are 40.3 kcal/mol and 41.2 kcal/mol respectively, which showed good agreement with the experimental results.”.

Fig. 2. Study on product stability. The rotation barrier and the $t_{1/2}$ of 3aa, 3aq, and 3ua.

(4) In line 153, the production rate of 3as is surprisingly high. Is there any effect of hydroxyl group here? Can the author give a reasonable explanation.

Response: Thanks for the kind suggestion. Indeed, the compound 3as exhibited a higher yield than 3aq and 3ar (as shown below). We hypothesize that this might be attributed to the reduced steric hindrance of alkyne coupling partner 2s, which possesses a shorter alkyl chain. Additionally, by analyzing the yields of 3ap, 3aq, and 3ar, we observed that the yield of annulation reaction decreased with an increase in the length of the internal alkyne chain. Moreover, the possibility of coordination between the hydroxyl group and the cobalt center cannot be ruled out at this stage.

(5) A related publication *Angew. Chem. Int. Ed.* 2023, 62, e202216863 should be added to the references.

Response: Thanks for the kind reminder. The corresponding review on the synthesis of C-C, C-N and N-N atropisomers (*Angew. Chem. Int. Ed.* 2023, 62, e202216863) has been included in ref. 19 in the revised manuscript, as shown below: 19. Lu, C.-J., Xu, Q., Feng, J. & Liu, R.-R. The Asymmetric Buchwald-Hartwig Amination Reaction. *Angew. Chem. Int. Ed.* **62**, e202216863 (2023).

(6) Are the unsymmetric non-terminal alkynes involved in the reaction regioselective? For example, how about the regioselectivity in **3at**?

Response: Thanks for the kind suggestion. Firstly, we have repeated the annulation with unsymmetric interterminal alkyne **2t** for the synthesis of **3at**. As a result, the conversion of reaction was not 100%, and the starting material of **1a** was recovered with a 45% isolated yield. By analyzing the NMR and the HPLC spectrum of product **3at**, we only detected one product with exclusive regioselectivity. And the description and Chemdraw has been modified in the revised manuscript, as shown below “Moreover, the aliphatic internal alkynes **2p–2s** also reacted smoothly to deliver *N–N* axially chiral isoquinolinones **3ap–3as** with 97–99% ee values. For the unsymmetrical internal alkyne **2t**, the reaction proceeded with exclusive regioselectivity to give **3at** in moderate yield and high enantioselectivity, and the starting material of **1a** was recovered with a 45% yield.”.

3at, 50%, 98% ee
recovery of **1a**, 45%

Re: Reviewer #2

We appreciate the generous support and kind comment of referee #2 on our work: “The novelty of protocol meets the publication request for the nature communication.”

Original comments: (1) Author should provide more applications to display the usefulness of the products instead of just providing a special example.

Response: We thank the editor for the encouraging comments, according to the constructive comments of reviewer, the further experimental explorations were conducted in order to further enhance the suitability of this protocol, and the synthetic utility of this developed protocol to access N-N atropisomers was further highlighted by post-functionalization. The oxidative product 5 could be obtained by treatment of N-N axially chiral isoquinolinone with DDQ in 82% yield at 120 °C for 2 h. Further treatment of 5 with *m*-CPBA afforded oxide 6, which was regarded as the chiral ligand to react with benzyl 2-diazo-2-phenylacetate and 1,2-dimethyl-1*H*-indole in the presence of palladium catalyst to give desired 10 in 90% yield and 58% ee value. As the experimental result shown, the carbonyl group of N-N axially chiral isoquinolinone could also be easily transformed into thiocarbonyl functionalities in 75% yield without detriment to enantioselectivities at the presence of Lawesson’s reagent. For detailed information, please refer to the part of figure 3 in our revised manuscript.

(2) Although author made some experiments to testify the C-H activation may be involved in the rate-determining step, how the chirality is controlled? what is the favorable model?

Response: Thank you for your helpful suggestion. Building upon the previous work on the cobalt-catalyzed annulation of benzamides with alkynes (ACIE, 2014, 53, 10209; JACS, 2018, 140, 7913; ACIE, 2018, 57, 2383), we could confirm that the reaction follows a $\text{Co}^{\text{II}}\text{-Co}^{\text{III}}\text{-Co}^{\text{I}}\text{-Co}^{\text{II}}$ catalytic cycle. Additionally, in our previous study on the Co/Salox-catalyzed C-H annulation of benzamides with alkynes to synthesize C-N atropisomers (Nat. Synth. 2022, 1, 709), we analyzed the potential configurations of the transition states involved in the reductive elimination process. And it was found that the C-N reductive elimination step played a crucial role in determining the stereochemistry. Therefore, for the annulation described in this paper, we propose that the C-N reductive elimination step may also be served as the stereo-determining step. And we have modified the description and the plausible catalytic mechanism of Fig 4d in revised manuscript, as shown below “*Based on the previous reports⁶⁵⁻⁷³ and the above results, a plausible mechanism was proposed (Fig. 4d). Cobalt salt (II) coordinates with L6 and AdCO₂H, and undergoes oxidation with O₂ to produce active Co(III)-species A, which then undergoes ligand exchange with substrate 1a to form the octahedral Co(III)-intermediate B. The C-H activation of B affords the key intermediate C. It is worth mentioning that another pathway involving the first C-H activation step and the following oxidation step to form the intermediate C could not be completely ruled out. Next, alkyne 2a coordinates with C and the ligand exchange occurred to form D. The migration insertion of alkyne into the C-Co bond affords the seven-membered alkenyl intermediate E. Finally, the reductive elimination of E leads to the formation of the N-N axially chiral product 3aa and meanwhile releases the Co(I) intermediate F. Based on our previous report for the synthesis of C-N axially chiral compounds,⁷⁰ the reductive elimination through transition states TS-I, which features the lowest-energy owing to the π - π stacking interactions between phenyl group of L6 with the 7-azaindole directing group of 1a, might serve as the enantio-determining step.*”.

(3) Reaction condition screening showed that AdCO_2H is very important for the yield improvement, author should provide the complete figure for the mechanism to exhibit the usefulness of AdCO_2H and the other key intermediates.

Response: Thank you for your kind reminder. The additive acid of AdCO_2H might serve as a coordinated ligand to stabilize the Co(III) intermediate and then facilitate the subsequent ligand exchange with alkyne. This effect has been observed in recently Co/Salox asymmetric catalysis reported by our group and Ackermann group (*Science*, 2023, 379, 1036; *Chem. Sci.*, 2023, 14, 7291), and the achiral cobalt catalysis previously (*Org. Lett.*, 2019, 21, 2863; *J. Org. Chem.*, 2020, 85, 4067). Moreover, we have added the effect of acid additives in the catalytic cycle and modified the catalytic mechanism and the related description, as shown in above record.

Re: Reviewer #3

We appreciate the generous support and kind comment of referee #3 on our work: *“Taking these into consideration, this reviewer would decide that the present work deserves publication in Nature Communications with some revision.”*

Original comments: (1) In the abstract, the authors wrote “Herein, the first catalytic synthesis of N-N axially chiral frameworks has been accomplished via...” . This sentence is ambiguous and misleading. The authors should more specifically elaborate what aspects of this work are different from the previous catalytic asymmetric synthesis of N-N atropisomers.

Response: **Thanks very much for the considerate suggestion. According to the suggestion, we have modified the description as “Herein, we report an efficient method for the enantioselective synthesis of N–N axially chiral frameworks via a cobalt-catalyzed atroposelective C–H activation/annulation process.” in the revised manuscript, and the revisions are highlighted in green background.**

(2) The present reaction is different from the previous annulation using 8-aminoquinoline-based directing group with respect to the skeleton of key cyclometalated intermediate (5,5-fused ring for this work and 5,6-fused ring for the previous work). Concerning this difference, the authors are suggested to address the following points: 1) Which is the better directing group for this annulation? A competition reaction under both the previous and present conditions would be informative. 2) The salox ligands with the phenyloxazoline moiety appear to show uniformly good enantioselectivity for this and previous reactions regardless of the presence of the bulky tert-butyl group (L1 vs L6, L7). Thus, the *tert*-Bu group seems to be increasing the catalytic activity while not particularly influence the enantioselectivity. What would be behind this effect? A discussion, or even insights from computational study would be worthwhile.

Response: **Thanks for the professional reminder.**

For the first comment, a competitive experiment by using 1a and 1a' bearing a 7-methyl-8-aminoquinoline directing group with a 1:1 ratio as substrate was conducted under optimized conditions as shown below. It was found the substrate of 1a and 1a' were completely transformed and the total yield was up to 99%. And the ¹H NMR spectrum suggested that the ratio of product 3aa and 3aa' was 1:1. This result demonstrated that both the N-amino-7-azaindole and 8-aminoquinoline are excellent directing groups for Co/Salox catalysis. And the result has been added

in the revised manuscript and SI as shown below (*Additionally, a competitive experiment by using 1a and 1a' bearing a 7-methyl-8-aminoquinoline directing group with a 1:1 ratio as substrate was conducted under optimized conditions, and the product 3aa and 3aa' were obtained with a total yield of 99% with a 1:1 ratio (See Table S10 in SI).*).

For the second comment, first we conducted experiments by using L1, L6, or L7 as chiral ligand under optimized conditions. The products 3aa were isolated with 36% (for L1), 91% (for L6), 87% (for L7) yields, respectively. Combined with the previous results in the preliminary optimization studies (15% yield for L1, 48% for L6, 42% for L7), we could find that the Salox ligands L6 and L7 bearing a bulky *tert*-butyl group indeed showed better catalytic activity than the L1. And this promoting effect has also been reported in the previous Co/Salox catalysis developed by Shi, Ackermann and our group (ACIE, 2023, e202218533; Science, 2023, 379, 1036; Chem. Sci., 2023, 14, 7291). Moreover, the Shi group proposed that the bulky *tert*-butyl group on the Salox ligand plays an important role in the coordination and migration insertion steps of the coupling partner and thus enables the high activity, regio-, and enantioselectivity. In our manuscript, we speculated that the chiral ligand L6 was also crucial in the catalytic cycle and thus facilitates the reaction. And a related description was added in the revised manuscript, as shown below “*And this promoting effect of the bulky ^tBu-Salox ligand has also been reported in the previous Co/Salox catalysis developed by Shi,⁶⁹ Ackermann⁵⁶ and our group⁷³.*”

(3) The way of writing in this manuscript is overly superfluous. For example, the first sentence of the main text, “The natural evolution of ecosystems for centuries...” has too little connection to what is actually achieved here. Also, the use of words like “bright future” in page 2 is too fluffy and objective and thus should be avoided. The authors should be aware of the scope/limitation of implications and impacts of their actual results and adopt a solid and subjective writing style. The present results are interesting enough

without superfluous writing.

Response: Thanks for the kind reminder. According to your comment, we have deleted the first sentence of “The natural evolution of ecosystems for centuries...”. Moreover, the inappropriate description about “bright future” in page 2 has been modified, as shown below: *“Very recently, our group,⁷⁰ and Shi’s group⁶⁵ independently developed an efficient chiral Co/Salox (salicyl-oxazoline) system, and demonstrated its excellent reactivity for assembling C–N and C–C axially chiral compounds. This strategy features significant advantages, making easily available cobalt salt catalyzed C–H functionalization for constructing axial chiral compounds being a promising research area. On the other hand, the nitrogen atoms in the N–N atropisomers may be derived from different substructures, giving rise to the possibility of forming diverse N–N axially chiral compounds. However, to the best of our knowledge, the construction of N–N axially chiral atropisomers via earth abundant transition-metal catalyzed C–H activation remains unprecedented.”*. The revisions are highlighted by green background in the revised manuscript.

Reviewers' Comments:

Reviewer #1:

Remarks to the Author:

The authors have done an excellent job addressing all of my concerns. I believe the authors have addressed this reviewer's concerns and the manuscript is ready for publication.

Reviewer #2:

Remarks to the Author:

Publication as it is.

Reviewer #3:

Remarks to the Author:

In the revised manuscript, the authors carefully addressed the points raised by this and other reviewers. Although the manuscript still needs significant language polishing, the scientific claims have been adequately supported. As such, this manuscript can be recommended for publication in Nature Communications.

NCOMMS-23-20080A-response letter

Detailed response to the Comments of Editorial Office:

Re: Reviewer #1

We appreciate a lot for the generous support from reviewer #1 on our work: “The authors have done an excellent job addressing all of my concerns. I believe the authors have addressed this reviewer's concerns and the manuscript is ready for publication”.

Re: Reviewer #2

We appreciate the generous support of referee #2 on our work: “Publication as it is”

Re: Reviewer #3

We appreciate the generous support and kind comment of referee #3 on our work: “In the revised manuscript, the authors carefully addressed the points raised by this and other reviewers. Although the manuscript still needs significant language polishing, the scientific claims have been adequately supported. As such, this manuscript can be recommended for publication in Nature Communications.”

Original comments: Although the manuscript still needs significant language polishing, the scientific claims have been adequately supported.

Response: Thanks very much for the considerate suggestion. According to the suggestion, we have modified some sentences and words in the revised manuscript, as shown below:

original description	changed to
Atropisomerism, as one kind of charming axial chirality, has greatly aroused explosive attention due to their fascinating architectures and	Atropisomerism, as one kind of intriguing axial chirality, has greatly aroused explosive attention due to their fascinating architectures and

excellent multidisciplinary application...	excellent multidisciplinary applications...
in preparation, separation, and purification methods and instruments	in preparation, separation, purification methods and instruments
This intriguing atropisomerism configurations have moved onto the research stage...	These intriguing atropisomerism configurations have moved onto the research stage...
which anomaly might be attributed to	which might be attributed to
the first-row 3d-metal cobalt-catalyzed C-H functionalization have gained...	the first-row 3d-metal cobalt-catalyzed C-H functionalization has gained...
which no doubtfully broaden the synthetic strategies...	which undoubtedly broaden the synthetic strategies...
the reaction was complete to furnish...	the reaction was able to produce...
conjugated enyne, and cyclopropyl acetylene were also compatible to this reaction	conjugated enyne, and cyclopropyl acetylene were also found to be compatible with this reaction
Subsequently, the oxidative product 5 could be...	In addition, the oxidative product 5 could be...
without detriment to enantioselectivities	without compromising enantioselectivity